# Bleeding Lesion from Roux-en-Y Hepaticojejunostomy: A Successful Combined Hemostasis with Dual Emission Laser 1.9/1.5 μm

**DOI:** 10.3390/diagnostics12092107

**Published:** 2022-08-30

**Authors:** Beatrice Marinoni, Luca Elli, Gian Eugenio Tontini, Lucia Scaramella, Roberto Penagini, Maurizio Vecchi, Nicoletta Nandi

**Affiliations:** 1Center for Prevention and Diagnosis of Celiac Disease, Gastroenterology and Endoscopy Unit, Foundation IRCCS Ca’ Granda Ospedale Maggiore Policlinico, Via F. Sforza 35, 20122 Milan, Italy; 2Department of Pathophysiology and Transplantation, University of Milan, 20122 Milan, Italy

**Keywords:** device-assisted enteroscopy, therapeutic intervention, liver transplantation, gastrointestinal bleeding, laser

## Abstract

A 28-year-old woman, with a history of liver transplantation with Roux-en-Y hepaticjejunostomy, was admitted for melena and severe anemia. Bidirectional endoscopy was normal. Capsule endoscopy demonstrated fresh blood in the efferent limb downstream of the jejuno-jejunostomy. Anterograde double-balloon enteroscopy (DBE) showed an adherent clot with a visible vessel oozing next to the hepaticojejunostomy. Bleeding was treated firstly with argon plasma coagulation and endoclips and further treated with dual emission laser, achieving complete hemostasis. At the 3 months follow-up, hemoglobin was stable without evidence of re-bleeding.

DBE is an effective and safe technique for managing complications in surgically altered anatomy [1]. Dual emission laser allows a precise hemostasis on the targeted mucosal surface, reducing the chance of unexpected injuries [2,3]. This case is the first describing a bleeding vessel in a liver-transplanted patient with Roux-en-Y hepaticojejunostomy treated by combining traditional endoscopic hemostatic techniques with an innovative one (dual emission laser). In particular, the bleeding source was in a very critical zone with the risk of damaging the anastomosis during cauterization and clip positioning. We think that the use of laser minimizes the possibility of uncontrolled cauterization and, thus, enables operating safely in difficult positions. In Figure 1 and Appendix A, the procedure is shown and explained in detail.

## Figures and Tables

**Figure 1 diagnostics-12-02107-f001:**
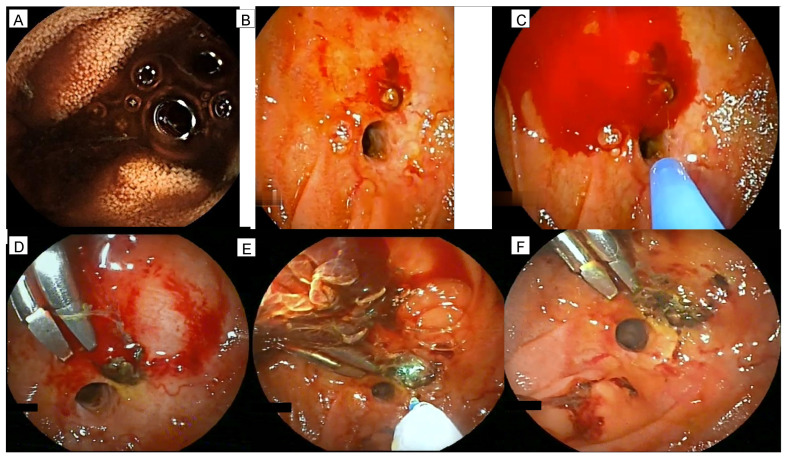
The main findings of the performed enteroscopies are shown. At capsule endoscopy, blood was present at jejunojejunostomy coming from the hepatic limb (**A**). During anterograde double-balloon enteroscopy, active bleeding from the afferent limb was observed with the presence of fresh blood at the jejunojejunostomy. Retrograde underwater exploration of the afferent limb showed the presence of an adherent clot close to the hepaticojejunostomy (**B**). Upon clot removal, an oozing hemorrhage from a visible vessel was observed and treated with argon plasma coagulation (30 W) (**C**). Subsequently, two endoclips were placed with a partial control of the bleeding (**D**). Finally, dual emission laser 1.9/1.5 μm was applied (Opera and Opera Evo by Quanta System, Samarate, Italy) (**E**) with the complete bleeding arrest (**F**).

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
