# Peer review of "Bleeding Lesion from Roux-en-Y Hepaticojejunostomy: A Successful Combined Hemostasis with Dual Emission Laser 1.9/1.5 μm"

_diagnostics, 2022, doi:10.3390/diagnostics12092107_

Round 1
Reviewer 1 Report
This is an interesting case in which successful bleeding control in the patient with liver transplantation with Roux-en-Y hepaticojejunostomy.
I have no comment on the contents, but "videocapsule enteroscopy" should be replaced with "capsule endoscopy", if it means the capsule endoscopy.
Author Response
Point 1: I have no comment on the contents, but "videocapsule enteroscopy" should be replaced with "capsule endoscopy", if it means the capsule endoscopy.
Response 1: With videocapsule enteroscopy, we meant capsule endoscopy. Therefore, accordingly, we replaced the term videocapsule enteroscopy with capsule endoscopy.
Reviewer 2 Report
Dear Author,
This is an interesting image case report. The followings are my comments.
#1. How to choose between APC and laser in this case ? Why not use more clip in this situation because the risk of perforation is a concern of Laser therapy in this situation
#2. What is the mechanism of bleeding in this case ?
Author Response
Point 1: How to choose between APC and laser in this case ? Why not use more clip in this situation because the risk of perforation is a concern of Laser therapy in this situation
Response 1: In this case, there was a well-defined and visible focal bleeding source that we were not able to completely arrest after using APC, thefore we decided to use laser because it has a superficial effect and it presents a more precise the cauterization. Thus, we were confident in not treating also the anastomosis which could have resulted in an anastomotic stricture. Finally, we did not use more clips because the tissue surrounding the anastomosis was fibrotic and hard at clipping; moreover, we were concerned about the possibility to clip part of the anastomosis or the distal part of the biliary tract.
Point 2: What is the mechanism of bleeding in this case ?
Response 2: The bleeding mechanism was a visible superficial vessel/vein just beside the anastomosis, initially covered by ad adherent clot.
Round 2
Reviewer 2 Report
The authors answer to the questions raised. I have no more questions.